# LLaVA-UHD v3: Progressive Visual Compression for Efficient Naive-Resolution Encoding in MLLMs

## Abstract

Visual encoding followed by token condensing has become the standard architectural paradigm in multi-modal large language models (MLLMs). Many recent MLLMs increasingly favor global naive-resolution visual encoding over slice-based methods. To investigate this trend, we systematically compare their behavior on vision-language understanding and attention patterns, revealing that global encoding enhances overall capability but at the expense of greater computational overhead. To address this issue, we present LLaVA-UHD v3, an MLLM centered upon our proposed Progressive Visual Compression (PVC) method, which can be seamlessly integrated into standard Vision Transformer (ViT) to enable efficient naive-resolution encoding. The PVC approach consists of two key modules: (i) refined patch embedding, which supports flexible patch-size scaling for fine-grained visual modeling, (ii) windowed token compression, hierarchically deployed across ViT layers to progressively aggregate local token representations. Jointly modulated by these two modules, a widely pretrained ViT can be reconfigured into an efficient architecture while largely preserving generality. Evaluated across extensive benchmarks, the transformed ViT, termed ViT-UHD, demonstrates competitive performance with MoonViT while reducing TTFT (time-to-first-token) by 2.4×, when developed within an identical MLLM architecture. Building upon ViT-UHD, LLaVA-UHD v3 also achieves competitive performance to Qwen2-VL, while further reducing TTFT by 1.9×. We will release all code and checkpoints to support future research on efficient MLLMs.

## 1 Introduction

Recent advances in Multimodal Large Language Models (MLLMs) (Wang et al. (2024); Team et al. (2025b); Li et al. (2024a); Wang et al. (2025b)) have significantly expanded the capabilities of vision-language understanding across diverse scenario, including optical character recognition (Hu et al. (2024); Lv et al. (2023)), remote sensing (Wang et al. (2025a); Kuckreja et al. (2024); Yao et al. (2025)), and mobile agents(Team et al. (2025a); Wu et al. (2024a)). To efficiently support such broad applicability, visual encoding followed by token condensing has become the general vision embedding paradigm in MLLMs, enabling more standardized and scalable multi-modal training.

This encode-then-compress framework, however, entails considerable computational overhead, as the vision encoder must process an excessive number of tokens before any reduction, which is exacerbated when image resolution increases (Yao et al. (2024b); Fan et al. (2024); Yao et al. (2024a)). Early MLLMs alleviated this problem through slice-based encoding (Li et al. (2024c); Zhang et al. (2024); Li et al. (2024d); Guo et al. (2024)), which encodes smaller image crops independently to reduce computation. However, this line of methods inevitably leads to a fragmented semantic context and limited global awareness. Previous studies (Huang et al. (2025); Yao et al. (2025)) manually design cross-slice interaction modules to supplement global information, yet these adapters are generally lack large-scale pretraining and so far have found limited adoption in industrial MLLMs.

By contrast, there is a growing trend toward adopting global naive-resolution encoding in recent state-of-the-art MLLMs (Wang et al. (2024); Liu et al. (2025b); Hong et al. (2025); Liu et al. (2024g)), which processes the entire image in a single forward pass. Although intuitively well-motivated, the principles behind this approach remain insufficiently explored.

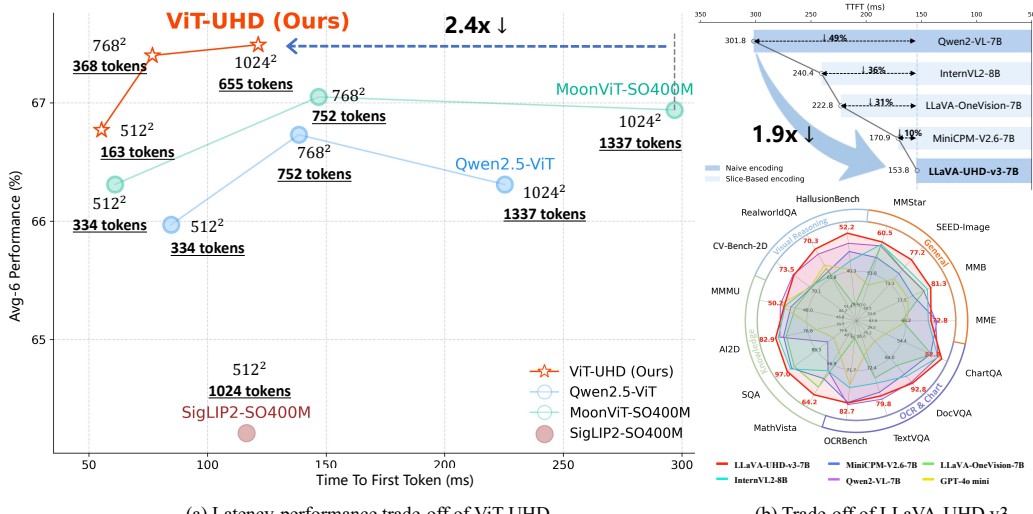

(a) Latency-performance trade-off of ViT-UHD

(b) Trade-off of LLaVA-UHD v3

Figure 1: ViT-UHD and LLaVA-UHD v3 exhibit a superior trade-off between efficiency and performance. (a) Within the LLaVA training paradigm, ViT-UHD achieves higher average performance across 6 benchmarks, such as MMBench and AI2D, compared to state-of-the-art vision encoders, while maintaining substantially greater computational efficiency (*e.g.*, achieving a 2.4× reduction in latency relative to MoonViT). (b) LLaVA-UHD v3 attains performance comparable to advanced MLLMs (*e.g.*, Qwen2-VL) across 15 diverse benchmarks, while delivering 1.9× efficiency gains.

In this work, we conduct controlled pilot experiments to investigate the mechanisms of global naive-resolution versus slice-based encoding in terms of spatial and semantic understanding. Moreover, we further analyze the internal patterns of attention activation. Our experiments indicate that global naive-resolution encoding yields stronger cross-modal understanding, yet its associated computational overhead remains substantial.

To address this issue, we introduce LLaVA-UHD v3, an MLLM built upon a novel Progressive Visual Compression (PVC) framework, which can be seamlessly integrated into standard ViT to enable efficient naive-resolution encoding. The PVC framework consists of two key components. **(i) Refined patch embedding (RPE).** As the tokenizer of images, the ViT patch-embedding layer fundamentally determines the granularity of visual tokenization. By scaling the patch size to finer levels via an equivalent weight-transformation scheme (Beyer et al. (2023)), this module supports more detailed visual modeling while preserving compatibility with pretrained ViTs. **(ii) Windowed token compression (WTC).** We insert lightweight, learnable local compressors within the encoder to progressively merge tokens inside local windows (*e.g.*, 2×2) across ViT layers, reducing sequence length during encoding. Initialized with average pooling and trained to produce content-adaptive weights, the module preserves local semantics while lowering ViT computation and LLM prefill cost. Jointly modulated by these two modules, a widely pretrained ViT can be reconfigured into a more efficient architecture, *e.g.*, via adjusting patch size, compressor count, and insertion indices.

Under the PVC framework, a widely pretrained ViT could be reconfigured into ViT-UHD. In the LLaVA setting (Guo et al. (2024); Li et al. (2024c)) with a Qwen2-7B (Team (2024b)) as the LLM, ViT-UHD achieves competitive performance across 6 benchmarks under varying input resolutions, while reducing time-to-first-token (TTFT) by up to 2.4× relative to MoonViT Team et al. (2025b), as shown in Fig. 1(a). (Please refer to Appendix A.1 for details.) Building on ViT-UHD, LLaVA-UHD v3 attains comparable performance on 15 vision–language benchmarks compared to Qwen2-VL, yet delivers 1.9× lower TTFT latency, shown in Fig. 1(b).

Our contributions are summarized as three manifolds. (1) We conduct controlled probes of global naive-resolution versus slice-based encoding, evidencing superior cross-modal understanding for the former and clarifying its underlying mechanisms. (2) We introduce a PVC framework, which jointly modulates refined patch embedding and windowed token compression to reconfigure widely pretrained ViTs into efficient naive-resolution ones. (3) Evaluations on extensive vision–language benchmarks demonstrates the effectiveness and efficiency of ViT-UHD and LLaVA-UHD v3.

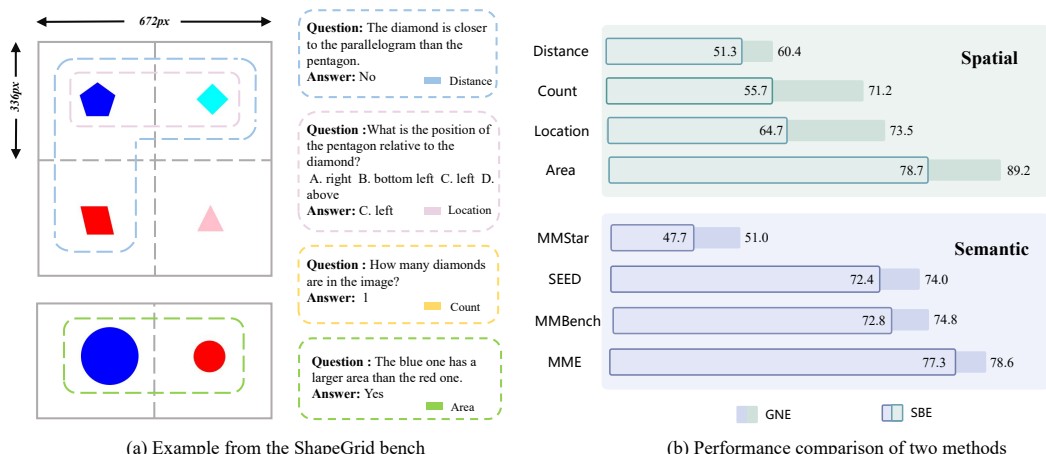

(a) Example from the ShapeGrid bench  (b) Performance comparison of two methods

Figure 2: ShapeGrid and model performance. (a) Examples from ShapeGrid bench, with each subset matched with color boxes. (b) Performance comparison between global naive-resolution encoding (GNE) and slice-based encoding (SBE) across different general benchmarks and ShapeGrid subsets.

## 2 PILOT EXPERIMENT

First, we start with a series of controlled pilot experiments that systematically contrast global naive-resolution visual encoding (GNE) and slice-based encoding (SBE) with respect to their capacities in semantic understanding and spatial perception. To guarantee experimental fairness, we adopt an identical model architecture, utilize an equivalent training corpus, and perform evaluation on a consistent suite of general-purpose benchmarks. Moreover, to decouple spatial perception from semantic understanding, we introduce a synthetic probe benchmark for spatial analysis.

### 2.1 EXPERIMENTAL PROTOCOL

**Probe benchmarking.** For evaluating general semantic understanding, we adopt a set of widely used multi-modal benchmarks, including MMBench, *etc.*, shown in Fig. 2(b). These benchmarks comprehensively cover diverse capabilities such as object attribution recognition, position relation recognition, optical character recognition (OCR), and visual reasoning. To assess spatial perception, it is important to note that the slicing operation intrinsic to slice-based visual encoding tends to fragment objects in natural images, which in turn introduces semantic discontinuities and makes it difficult to disentangle spatial perception from semantic understanding. To solve this problem, we construct a controlled synthetic dataset, ShapeGrid. The ShapeGrid bench is generated from a template pool of parameterized geometric shapes, incorporating controlled variations in color, scale, and position. Final images are composed using predefined layouts aligned with slice-based encoding strategies. The dataset supports four spatial perception tasks as shown in Fig. 2(a). Construction details are detailed in the Appendix A.2.2.

**Model configuration and training setup.** We adopt a standard MLLM architecture with SigLIP2-SO400M (Tschannen et al. (2025)) as the vision encoder, a pixel-unshuffle (Dong et al. (2024)) projector, and Qwen2-7B (Team (2024b)) as the LLM. Training follows a two-stage paradigm (Zhang et al. (2024)), first optimizing the vision encoder and projector while freezing the LLM, then fine-tuning all parameters. Both GNE and SBE methods are configured with a maximum resolution of 1008×1008 pixels. Details are provided in Appendix A.2.1.

### 2.2 EXPERIMENTAL ANALYSIS

**Overall performance.** In Fig. 2(b), GNE method achieves significantly higher performance than SBE method, with clear improvements observed on both the general semantic benchmarks and the ShapeGrid. On average, it improves general semantic understanding by 2.1% (69.6% vs. 67.5%) and spatial perception by 11.0% (73.6% vs. 62.6%), indicating that preserving holistic visual context consistently benefits both semantic understanding and spatial perception.

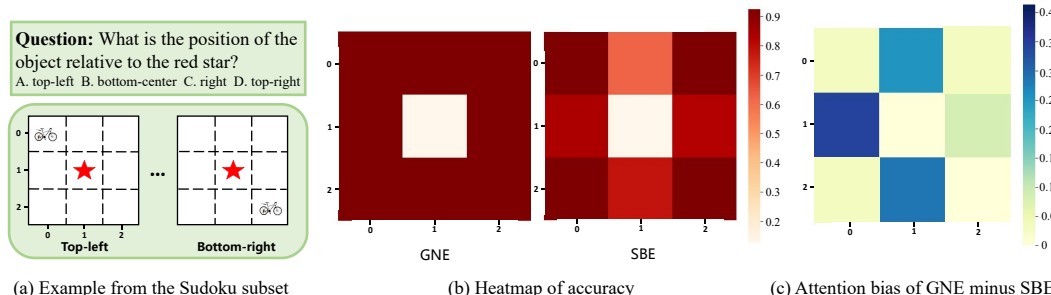

(a) Example from the Sudoku subset     (b) Heatmap of accuracy     (c) Attention bias of GNE minus SBE

Figure 3: Illustration and analysis on ShapeGrid-Sudoku subset. (a) Example from the Sudoku subset. (b) Accuracy heatmap of models with global naive-resolution encoding (GNE) vs. slice-based encoding (SBE) on the Sudoku subset. (c) Attention score bias map showing the difference in attention activation between GNE and SBE.

**Behavior on spatial directional perception.** To better evaluate spatial positional understanding, we extend the original location subset into a Sudoku-style variant, where directional localization is explicitly isolated for clear evaluation. As for data construction, each image follows a $3\times3$ layout with a fixed central anchor and surrounding objects sampled from the template pool and additional real-world categories, which contains 8,000 image-query pairs focused on relative direction prediction, as shown in Fig. 3(a). Further details are provided in the Appendix A.2.2. As shown in Fig. 3(b), GNE method yields uniformly high accuracy on Sudoku subset across all directions, reflecting balanced spatial understanding. In contrast, SBE method suffers systematic degradation along the vertical and horizontal axes, forming a clear cross-shaped bias. These results reveal an inherent systematic flaws in SBE approach. The trend that GNE alleviates such flaws more effectively can also be observed when comparing Qwen2.5-VL (GNE) and MiniCPM-o 2.6 (SBE), as shown in Appendix A.2.3.

**Attention pattern analysis.** To further investigate this phenomenon, we analyze the attention activation patterns of GNE and SBE methods. Specifically, we compute the average attention from answer text tokens to image tokens within the ground-truth cell and visualize the differences in Fig. 3(c). The results reveal a clear anisotropy in SBE: attention to the top, bottom, left, and right positions is markedly weaker, whereas the four corners exhibit similar activation levels. This suggests that image partitioning in SBE disrupts spatial uniformity, introducing a systematic directional bias. In contrast, GNE maintains a more evenly distributed and coherent attention pattern, thereby preserving holistic spatial relationships and enabling more accurate localization.

### 2.3 Conclusions on Pilot Experiment

In summary, we demonstrate that global naive-resolution encoding consistently outperforms slice-based methods across both general semantic benchmarks and spatial perception probes. The superiority is particularly pronounced in directional localization, where global encoding eliminates the cross-shaped bias inherent in slice-based methods and maintains balanced attention distributions. These findings indicate that global encoding not only enhances overall capability in vision-language understanding but also provides mechanistic insights into its advantage in spatial reasoning. Nevertheless, global encoding still incurs substantial computational cost, as discussed in previous work (Yao et al. (2024b); Liu et al. (2021); Li et al. (2022)), underscoring the urgent need for a naive and efficient visual encoding paradigm.

## 3 LLaVA-UHD v3

Building upon conclusions of pilot experiments, we propose LLaVA-UHD v3, an MLLM equipped with our Progressive Visual Compression (PVC) approach for efficient naive-resolution encoding. LLaVA-UHD v3 follows the standard MLLM architecture of a vision encoder, a projector, and a large language model (LLM), as illustrated in Fig. 4. PVC framework is applied within the vision encoder, where it integrates two modules: Refined patch embedding (RPE) and Windowed token compression (WTC). By progressively condensing tokens while preserving global contextual infor-

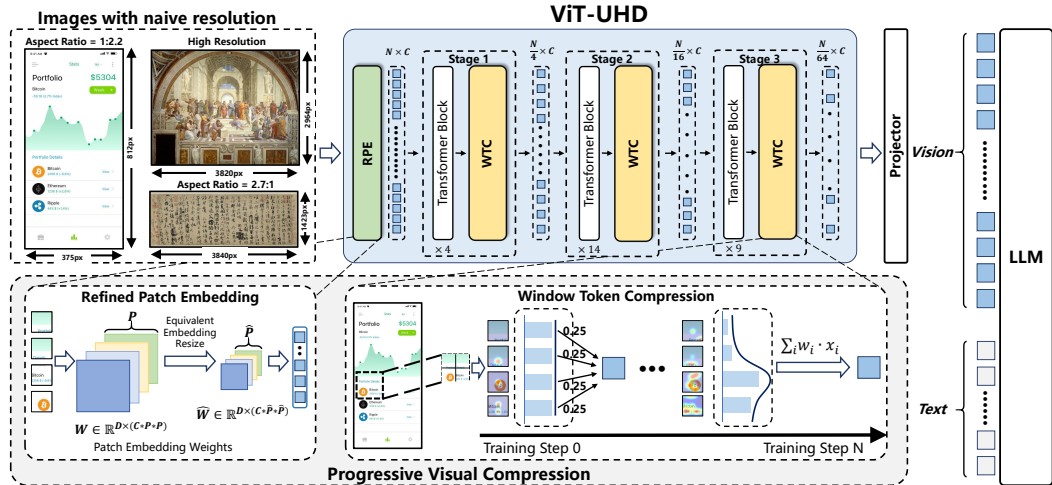

Figure 4: Overview of the architecture of LLaVA-UHD v3. ViT-UHD first utilizes Refined Patch Embedding (RPE) to tokenize images with naive-resolution into fine-grained tokens. Window Token Compression (WTC) modules are inserted at multiple stages to progressively reduce token length while learning local semantics. The final vision tokens are then projected into the LLM.

mation, these components reconfigure a pretrained ViT into an efficient and generalizable model, referred to as ViT-UHD.

## 3.1 ViT-UHD

ViTs are built upon the transformer architecture, where self-attention serves as the core operation (Vaswani et al. (2017)). Although this design offers strong representational capacity, a key limitation lies in its quadratic complexity with respect to token length. Higher input resolutions substantially increase visual token counts, resulting in elevated memory demands and inference latency during both visual encoding and LLM pre-filling. A more fundamental alternative is to replace self-attention with linear variants (Choromanski et al. (2020)) or RNN-style architectures (Yang et al. (2024); Peng et al. (2023)). However, such designs typically require large-scale training from scratch and fail to inherit the well-established modeling capacity of pretrained transformer-based models. Therefore, the core idea of our approach is to keep the token length within a controllable range while enriching fine-grained visual features. To achieve this, we apply RPE to flexibly expand visual granularity, and then employ WTC to progressively condense tokens and reduce redundancy.

### 3.1.1 REFINED PATCH EMBEDDING

Generally, the objective of this module is to reduce the patch size $P$ employed in patch embedding (e.g., from $14 \times 14$ to $10 \times 10$), which increases the modeling granularity of visual tokens, while preserving the representational equivalence of the original embeddings. Specifically, for an original patch embedding kernel weights $W \in \mathbb{R}^{D \times (C*P*P)}$, where $C$ denotes the number of input channels (e.g., 3 for RGB images), and $D$ the embedding dimension, we define the transformed weights $\hat{W} \in \mathbb{R}^{D \times (C*\hat{P}*\hat{P})}$, where $\hat{P} < P$. Given a patch vector of the image $t \in \mathbb{R}^{1 \times (C*P*P)}$ and its finer-level counterpart $\hat{t} \in \mathbb{R}^{1 \times (C*\hat{P}*\hat{P})}$, we can derive a linear transformation matrix $B \in \mathbb{R}^{(C*P*P) \times (C*\hat{P}*\hat{P})}$ that maps the coarse patch to the finer one, i.e., $\hat{t} = tB$. To ensure representational equivalence between the original patch embedding $tW^\top \in \mathbb{R}^{1 \times D}$ and refined patch embedding $\hat{t}\hat{W}^\top \in \mathbb{R}^{1 \times D}$, we require $tW^\top \approx \hat{t}\hat{W}^\top$. Since $\hat{t} = tB$, this condition can be rewritten as $B\hat{W}^\top \approx W^\top$. Following previous work (Beyer et al. (2023)), we utilize the least-squares solution to this relation and yields $\hat{W} = (B^\top)^+ W$, where $(B^\top)^+$ denotes the pseudo-inverse of $B^\top$. Through this weight transformation, we update a patch embedding weights $W$ into new one $\hat{W}$, which could encode each finer-level patch vector into an image token.

So that, given an input image $I$ of size $H_I \times W_I$, patch embedding with transformed weights $\hat{W} \in \mathbb{R}^{D \times (C*\hat{P}*\hat{P})}$ produces a token feature map of size $h_I \times w_I = (H_I/\hat{P}) \times (W_I/\hat{P})$, which can be

flattened into $N$ visual tokens $\{x_i = t_i W^\top\}_{i=1}^N$, where $N = h_I \times w_I$, $t_i \in \mathbb{R}^{1 \times (C * \hat{P} * \hat{P})}$ is $i$-th image patch vector.

### 3.1.2 WINDOWED TOKEN COMPRESSION

Following patch embedding, a large number of visual tokens are forwarded into transformer layers for further encoding. Especially, when the patch size decreases using RPE, the number of visual tokens increases quadratically, leading to heavy computational overhead. To mitigate the burden on the ViT backbone as well as the pre-filling stage of the LLM, we insert a set of lightweight compression layers within the ViT. These layers hierarchically reduce the token sequence length, thereby partitioning the backbone into $J$ stages with progressive compression, as shown in Fig. 4.

**Plain pooling strategy.** In the $j$-th compression stage, we predefine a local window of size $2 \times 2$ and partition the entire feature map into non-overlapping windows. Within each window, an average pooling strategy is applied to aggregate the tokens as

$$x_{avg} = \frac{1}{2 \times 2} \sum_{i=1}^{2 \times 2} x_i, \tag{1}$$

so that after the $j$-th compression stage, the token feature map resolution is downsampled by $2\times$ and the number of visual tokens is condensed to $\{x_i\}_{i=1}^{N/4^j}$. Although this simple pooling operation assigns uniform weights to all tokens and ignores their semantic importance, our experiments show that it facilitates more stable and effective convergence. In contrast, parameterized methods such as pixel-unshuffle (Wang et al. (2025b)) intuitively provide enhanced local semantic modeling capacity but exhibit convergence challenges when inserted in early ViT layer (see Appendix A.5.1).

**Enhanced pooling mechanism.** To enhance the expressiveness of pooling operation while maintaining efficient convergence, we introduce a content-adaptive pooling mechanism. Specifically, it first performs feature average pooling to obtain an aggregated token $x_{avg} \in \mathbb{R}^{1 \times D}$ as mentioned in Equ. 1. This token is then concatenated to each of the tokens $\{x_i\}_{i=1}^4$ within the local window to form extended representations $\hat{x}_i = [x_i; x_{avg}] \in \mathbb{R}^{1 \times 2D}$, followed by passing through an MLP ($f_\theta : \mathbb{R}^{1 \times 2D} \to \mathbb{R}^{1 \times D}$) to compute channel-wise attention logits $a_i = f_\theta(\hat{x}_i)$. In the end, the Equ. 1 can be improved as a learnable weighted aggregation like

$$x_{avg} = \sum_{i=1}^{2 \times 2} w_i x_i = \sum_{i=1}^4 \frac{\exp(a_i)}{\sum_{i=1}^4 \exp(a_i)} x_i. \tag{2}$$

During early training, attention weights $w_i$ tend to be uniform by zero initializing the MLP $f_\theta$, which effectively approximates the average pooling for stabilize optimization. As training progresses, $f_\theta$ gradually learns semantically meaningful aggregation weights, allowing the model to preserve key spatial and semantic structures while reducing the token count.

### 3.1.3 JOINT MODULATION

Building on the two proposed modules, the PVC framework reconfigures a pretrained ViT by jointly modulating key structural factors. The patch size $P$ in refined patch embedding controls token granularity, while the number of $J$ WTC and their positions $j$ in ViT layers govern hierarchical token reduction. Following established design principles (Liu et al. (2021; 2023a)), these factors systematically transform a naive-resolution ViT into a more efficient architecture.

## 3.2 VISION PROJECTOR AND LLM

**MLP projector**. Considering that ViT-UHD already integrates hierarchical token compression within the vision encoder, the role of the projector is simplified to the pure feature alignment. We concretely employ a simple MLP like (Li et al. (2024c;a)) to directly map the visual tokens into the language embedding space. Such a design avoids redundant manual engineering and offers a more standardized and scalable solution, facilitating large-scale MLLM training.

**Large language model**. Compared with slice-based methods such as (Guo et al. (2024); Wang et al. (2025b); Yao et al. (2024b)), our design encodes images at naive-resolution and thus obviates the need for inserting special tokens (*e.g.*, ";" and "/n") to explicitly inform the LLM of slice layouts. Instead, we simply adopt the `<image>` and `</image>` placeholders to delimit the visual context, allowing all tokens to be seamlessly injected into the language model as contextual inputs.

| Config | Stage 1 | Stage 2 | Stage 3 |
|---|---|---|---|
| Data type | Image-text pairs, OCR caption | OCR/Doc/Chart/detailed captions, interleaved data, pure text corpus | Instruction-following data, reasoning data, pure text SFT data |
| Data volume & Batch size | 4.3M, 256 | 5M, 128 | 13.3M, 128 |
| Trainable Parts & LR | ViT-UHD: 1e-5 Projector: 2e-4 | ViT-UHD/Projector: 1e-5 LLM: 2e-5 | ViT-UHD/Projector: 1e-5 LLM: 2e-5 |

Table 1: Training configuration for LLaVA-UHD v3.

## 4 EXPERIMENTS

In this section, we introduce the training recipe, detail the experiment setting , analyze the model performance and further conduct the ablation study.

### 4.1 TRAINING RECIPE

**Overall training strategy.** We adopt a three-stage training paradigm to gradually enhance the capabilities of LLaVA-UHD v3, as shown in Tab. 1. Stage 1 focuses on vision-language alignment, where a plain ViT initialized from MoonViT-SO-400M is transformed into ViT-UHD and aligned with a Qwen2-7B LLM via an MLP projector. Stage 2 conducts joint multi-modal pre-training, aligning vision and language features in a unified space. Stage 3 applies supervised fine-tuning with instruction and reasoning dataGuo et al. (2025) for coherent response generation. The overall training period is ∼300h with 32× 80G-A100 GPUs. Details are provided in Appendix A.3.

**Ablation training strategy**. The MLLM uses SigLIP2-SO-400M as the vision encoder, an MLP projector, and Qwen2-7B as the LLM, with training and inference fixed at a resolution of 1024×1024. We follow the LLaVA pipeline, using LLaVA-Pretrain-558K for vision-language alignment and SFT-858K Zhang et al. (2024) for supervised fine-tuning. For the baseline, we insert a WTC layer using pixel-unshuffle in the last ViT layer. For the PVC incremental setting, an extra ViT pre-alignment stage is added before standard LLaVA training to mitigate the perturbation of ViT feature modeling introduced by PVC. The pre-alignment data is a subset of stage 1 in Tab. 1. More details are supplied in Appendix A.4.

### 4.2 EXPERIMENT SETTING

**Benchmarks**. We evaluate our model on a comprehensive suite of multi-modal benchmarks, which are divided into four categories: (1) General benchmarks including MME (Fu et al. (2024)), MMB (Liu et al. (2024d)), SEED (Li et al. (2024b)) and MMStar (Chen et al. (2024a)). (2) Knowledge benchmarks such as SQA (Lu et al. (2022)), MMMU (Yue et al. (2024)), AI2D (Kembhavi et al. (2016)) and MathVista (Lu et al. (2024)). (3) OCR&Chart benchmarks consist of OCR-Bench (Liu et al. (2024f)), TextVQA (Singh et al. (2019)), DocVQA (Mathew et al. (2021)) and ChartQA (Masry et al. (2022)). (4) Visual reasoning benchmarks like HallusionBench (Guan et al. (2024)), RealWorldQA (XAI (2024)) and CV-Bench (Tong et al. (2024)).

**Evaluation Metrics**. (1) We present the amount of training image-text pairs (#Data) for fair comparison. For methods that reports the number of training tokens (#token), we roughly compute the number of image-text pairs by dividing #token by a standard maximum token limitation (*i.e.*, 4096) in a single batch. (2) We compute the compression ratio as the final LLM token count divided by the number of tokens after the ViT patch embedding, where a higher value indicates denser visual information and more efficient inference. (3) We report Time-to-First-Token (TTFT) under a fixed resolution of $1024 \times 1024$, averaged across 100 inferences, using FlashAttention2 (Dao (2023)) and bf16 data type on one single 80G-A100 GPU for the whole model. (4) In ablation study, we report the average accuracy of benchmarks within each sub-category.

**Counterparts.** The compared models are categorized as three manifold. (1) Commercial open-source MLLMs, trained large corpora (*e.g.* over 100M vision-language samples) by industry labs, which includes MiniCPM-V2.6 (Yao et al. (2024b)), Qwen2-VL (Wang et al. (2024)), DeepSeek-VL2 (Wu et al. (2024b)), InternVL2 (Chen et al. (2024b)), POINTS (Liu et al. (2024e)), VILA[2] (Fang et al. (2024)) and *etc.* (2) Academic-scale MLLMs like LLaVA-OneVision (Li et al. (2024a)), Cambrian-1 (Tong et al. (2024)), FastVLM (Vasu et al. (2025)), Valley2 (Wu et al. (2025))

| Model | LLM | #Data | Ratio | General | | | Knowledge | | | | |
|---|---|---|---|---|---|---|---|---|---|---|---|
| | | | | MME | MMB | SEED-Image | MMStar | MMMU | AI2D | SQA | MathVista |
| **Academic Open-source MLLMs** | | | | | | | | | | | |
| FastVLM | Vicuna-7B | 1.6M | **64** | - | - | - | - | 37.3 | - | 74.1 | - |
| VILA[2] | Llama3-8B | 57.5M | 1 | - | 76.6 | 66.1 | - | 40.8 | - | 87.6 | - |
| LLaVA-OneVision | Qwen2-7B | 9.4M | 4 | 1998.0 | 80.8 | 75.4 | **61.7** | 48.8 | 81.4 | 96.0 | 63.2 |
| Cambrain-1 | Llama-3-8B | 9.5M | 16 | - | 75.9 | 74.7 | - | 42.7 | 73.0 | 80.4 | 49.0 |
| Valley2 | Qwen2.5-7B | 27.2M | 4 | - | 80.7 | - | 61.0 | **57.0** | 82.5 | - | **64.6** |
| **Closed-source MLLMs** | | | | | | | | | | | |
| GPT-4o-mini | - | - | - | 2041.0* | 76.8* | 72.5* | 50.0* | 54.1* | 73.7* | 82.6* | 52.5* |
| Gemini-1.5-Pro | - | - | - | - | 73.9 | - | 59.1 | 60.6 | 79.1 | - | 58.3 |
| Step-1.5V-mini | - | - | - | - | 79.7 | - | 54.7 | 51.7 | 81.3 | - | 57.8 |
| **Commercial Open-source MLLMs** | | | | | | | | | | | |
| InternVL2 | InternLM2.5-7B | ~100M | 4 | 2210.3 | 79.5 | 76.2* | 60.7 | 54.1 | 83.0 | 84.6 | 58.3 |
| POINTS | Qwen-2.5-7B | 431.0M | 4 | 2195.2 | **83.2** | 74.8 | 61.0 | 49.4 | 80.9 | 94.8 | 63.1 |
| MiniCPM-V-2.6 | Qwen-7B | 460.0M | 16 | **2348.4** | 78.0 | 74.1 | 57.5 | 49.8 | 82.1 | 96.9* | 60.6 |
| Qwen2-VL | Qwen2-7B | ~700.0M | 4 | 2326.8 | 80.7 | 75.3* | 60.7 | 54.1 | 83.0 | 84.6* | 58.2 |
| DeepSeek-VL2 | DeepSeekMoE-16B | ~203.0M | 4 | 2123.0 | 79.9 | 76.8* | 57.5 | 49.7 | 81.7 | 96.2* | 61.9 |
| LLaVA-UHD-v3 (ours) | Qwen2-7B | 20.1M | **64** | 2183.6 | 81.3 | **77.2** | 60.5 | 50.2 | 82.9 | 97.0 | 64.2 |

Table 2: Performance comparison of LLaVA-UHD v3 and state-of-the-art MLLMs on general and knowledge benchmarks. The performance is reported from its technical paper or OpenCompass leaderboard. "*" indicates the performance we reproduce using VLMEval-Kit. "Ratio" denotes the compression ratio defined in Sec.4.2.

| Model | LLM | #Data | Ratio | Visual Reasoning | | | OCR & Chart | | | |
|---|---|---|---|---|---|---|---|---|---|---|
| | | | | HallusionBench | RealworldQA | CV-Bench-2D | OCRBench | TextVQA | DocVQA | ChartQA |
| **Academic Open-source MLLMs** | | | | | | | | | | |
| FastVLM | Vicuna-7B | 1.6M | **64** | - | - | - | - | 67.4 | 62.8 | - |
| VILA[2] | Llama3-8B | 57.5M | 1 | - | - | - | - | 73.4 | - | - |
| LLaVA-OneVision | Qwen2-7B | 9.4M | 4 | 31.6 | 66.3 | 69.7* | 62.2 | 75.9* | 87.5 | 80.0 |
| Cambrain-1 | Llama-3-8B | 9.5M | 16 | - | 64.2 | - | 62.4 | 71.7 | 77.8 | 73.3 |
| Valley2 | Qwen2.5-7B | 27.2M | 4 | 48.0 | - | - | 84.2 | - | - | - |
| **Closed-source MLLMs** | | | | | | | | | | |
| GPT-4o-mini | - | - | - | 42.5* | 67.2* | 69.9* | 78.5* | 68.0* | 78.2* | 29.04* |
| Gemini-1.5-Pro | - | - | - | 45.6 | - | - | 75.4 | - | - | - |
| Step-1.5V-mini | - | - | - | 46.7 | - | - | 77.3 | - | - | - |
| **Commercial Open-source MLLMs** | | | | | | | | | | |
| InternVL2 | InternLM2.5-7B | ~100M | 4 | 45.2 | 64.4 | 70.1* | 79.4 | 77.4 | 91.6 | 83.3 |
| POINTS | Qwen-2.5-7B | 431.0M | 4 | 46.0 | 67.3 | - | 72.0 | - | - | - |
| MiniCPM-V-2.6 | Llama-3-8B | 460.0M | 16 | 48.1 | 65.5* | 69.7* | 85.2 | 80.1 | 90.8 | 79.4* |
| Qwen2-VL | Qwen2-7B | ~700.0M | 4 | 50.6 | 70.1 | **76.0*** | 86.6 | 84.3 | **94.5** | 83.0 |
| DeepSeek-VL2 | DeepSeekMoE-16B | ~203.0M | 4 | 43.8 | 65.4 | - | 83.2 | 83.4 | 92.3 | 84.5 |
| LLaVA-UHD-v3 (ours) | Qwen2-7B | 20.1M | **64** | 52.2 | 70.3 | 73.5 | 82.7 | 79.8 | 92.8 | 82.8 |

Table 3: Performance comparison of LLaVA-UHD v3 and state-of-the-art MLLMs on OCR&Chart and visual reasoning benchmarks.

and *etc.* (3) Closed-source MLLMs such as GPT-4o-mini (OpenAI (2024)), Gemini-1.5-Pro (Team (2024a)) and Step-1.5V-mini (Team (2025)).

### 4.3 MODEL PERFORMANCE

In Tab. 2, compared with commercial open-source MLLMs, our LLaVA-UHD v3 achieves strong performance despite using one order of magnitude less training data than Qwen2-VL ( 20.1M vs. 700.0M). On general benchmarks, it demonstrates clear advantages, particularly on MMB (81.3 vs. 80.7) and SEED-Image (77.2 vs 75.3), showing the effectiveness of our approach in general multimodal understanding. On knowledge benchmarks, especially MathVista, our model surpasses both InternVL2 and MiniCPM-V-2.6, highlighting the superior capability of our global encoding mechanism in capturing complex geometric and spatial relationships.

Visual reasoning and OCR&Chart tasks require more fine-grained visual representations. As illustrated in Tab. 3, although our LLaVA-UHD v3 has the highest compression ratio (64), it significantly outperforms the slice-based encoding methods MiniCPM-V-2.6 (ratio = 16) and Intern-VL-2 (ratio = 4) on spatial reasoning datasets such as HallusionBench and CV-Bench. Moreover, on the OCR&Chart subset, our MLLM delivers comparable performance compared to Qwen2-VL and MiniCPM-V-2.6, demonstrating that our high-compression global encoding not only preserves but also enhances fine-grained visual understanding required for text-rich and structured visual tasks.

### 4.4 ABLATION STUDY

To better understand the contributions of the proposed components in ViT-UHD and the best modulation configuration, we conduct a comprehensive ablation study, some of which are in Appendix A.5.

| Method | Patch Size | # WTC | Layer Id | Tokens↓ | TTFT (ms)↓ | Avg↑ | General↑ | OCR&Chat↑ | Visual Reasoning↑ | Knowledge↑ |
|---|---|---|---|---|---|---|---|---|---|---|
| Baseline | 16 | 1 | 27 | 1024 | 233 | 62.1 | 69.8 | 61.8 | **55.7** | 59.7 |
| + WTC | | | | | | | | | | |
|   - Avg-pooling | 16 | 2 | 4,18 | 256 | 82 | 59.2 | 68.9 | 53.4 | 53.2 | 59.6 |
|   - CA-pooling | 16 | 2 | 4,18 | 256 | 83 | 60.7 | **70.5** | 55.6 | 55.2 | **59.9** |
|   - Pixel-unshuffle | 16 | 2 | 4,18 | 256 | 83 | 57.5 | 64.9 | 53.5 | 52.9 | 57.5 |
| + RPE | 8 | 3 | 4,18,27 | 256 | 160 | **63.0** | 70.3 | **64.6** | 55.6 | 59.8 |

Table 4: Evaluation on proposed modules including refined patch embedding (RPE) and windowed token compression (WTC). "# WTC" denotes the number of inserted WTC in ViT, "CA pooling" the brief of content-adaptive pooling described in Sec. 3.1.2. For all configurations, the WTC inserted at the 27-th layer is implemented using pixel-unshuffle.

**Module ablations**. As shown in Tab. 4, introducing WTC using average pooling substantially improves inference efficiency (from 233 ms to 82 ms). However, the sharp token reduction (1024 to 256) weakens fine-grained representation, leading to severe OCR&Chart degradation. In contrast, our content-adaptive pooling alleviates this issue and improves performance on the general subset, highlighting the importance of dynamically modeling local semantics, while incurring nearly no extra cost compared to average pooling. Utilizing a parameterized module like pixel-unshuffle as WTC causes notable performance degradation due to convergence challenges. Combining RPE with 3 WTC layers yields a favorable trade-off, delivering 1.5× higher efficiency (160 ms vs. 233 ms) and boosting average accuracy (63.0 vs. 62.1) across all benchmarks.

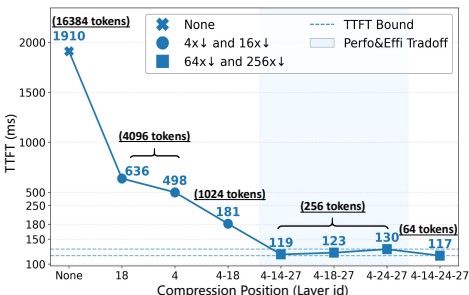

Figure 5: Efficiency of varying compression position. Optimal trade-off are shown in the light blue region. 4× denotes a different compression ratio.

**Joint modulation for superior trade-off.** We evaluate the impact of different configuration of WTC on the TTFT latency of LLaVA-UHD v3 under a patch size of $P = 8$ and an input resolution of $1024 \times 1024$. As shown in Fig. 5, adding 1 or 2 WTC layers significantly improves efficiency, with earlier inserted layers yielding greater latency reduction. Interestingly, when 3 WTC layers are introduced, the efficiency gain reaches saturation, and further increasing the number of layers or adjusting their positions brings negligible benefit.

## 5 RELATED WORKS

**Sliced-based Visual Encoding.** Slice-based encoding improves efficiency by dividing high-resolution images into smaller slices, processing each independently with a ViT, and then concatenating their features (Liu et al. (2024b); Zhai et al. (2023); Liu et al. (2024a); Huang et al. (2025)). LLaVA-UHD (Guo et al. (2024)) computes an optimal slicing strategy while preserving aspect ratios, and SPHINX (Liu et al. (2024a)) reduces sequence length by padding partial slices with "/n" tokens. **Naive-Resolution Image Encoding.** Naive-resolution encoding directly processes images at their original resolution without resizing (Dehghani et al. (2023)), preserving holistic spatial information. Recent works such as Qwen2.5-VL (Bai et al. (2025)) and MiMo-VL (Team et al. (2025a)) improve this approach through window attention mechanism and position embedding like 2d-RoPE. **Visual Feature Compression.** To handle the long token sequences output by ViTs, projectors are used for feature compression and alignment. Common designs include MLPs (Liu et al. (2023b)), Q-Formers (Li et al. (2023)), and Re-samplers (Barr et al. (2022); Bai et al. (2023)), with recent variants exploring pooling and attention-based compression (Cha et al. (2024); Liu et al. (2025a)).

## 6 CONCLUSION

We introduced LLaVA-UHD v3, an MLLM built upon Progressive Visual Compression (PVC) for efficient naive-resolution encoding. By integrating refined patch embedding and windowed token compression, a standard ViT will be reconfigured into an efficient yet generalizable encoder. Extensive evaluations demonstrate competitive performance and significantly reduced inference latency compared with SoTA ViTs and MLLMs.

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

USE OF LARGE LANGUAGE MODELS

We acknowledge the assistance of advanced large language models like OpenAI GPT5 in improving the clarity and conciseness of the manuscript. Their support in language refinement contributed to enhancing the overall presentation of this paper.

# A  APPENDIX

In this appendix, we provide comprehensive supplementary materials to support the main paper. We first present the detailed model architecture illustrated in Fig. 1, followed by a description of the training data, evaluation metrics, and result analyses. We then elaborate on the construction and experimental setup of the pilot experiment, as well as the complete training dataset composition and training details of the overall LLaVA-UHD v3 training. Additional experiments are further reported to validate the effectiveness of the proposed method, and finally, we include future work and case studies for qualitative analysis.

## A.1  TRAINGING SETTING FOR FIG.1

### A.1.1  COMPARISON OF LATENCY–PERFORMANCE TRADE-OFF ACROSS VISION BACKBONES

**A.1.1.1 Model Configuration**

We compare the performance of our ViT-UHD with recent open-source state-of-the-art vision encoders like Qwen2.5-ViT, MoonViT-SO400M, and SigLIP2-SO400M in MLLMs scenario. To balance efficiency and performance, we adopt different projectors for different vision encoders. Specifically, ViT-UHD and SigLIP2-SO400M adopt an MLP projector, while Qwen2.5-ViT and MoonViT-SO400M use pixel-unshuffle projector. All MLLMs adopt the same LLM (Qwen2-7B).

- SigLIP2-SO400M: A vision encoder pre-trained using a mixture of objectives, including image-text contrastive learning, self-masked image modeling, and auto-regressive language modeling. The patch size is 16.

- Qwen2.5-ViT: The pre-trained vision encoder of Qwen2.5-VL Bai et al. (2025). Qwen2.5-ViT is a naive-resolution ViT that replaces most global attention layers with window attention to improve efficiency, retaining only 4 global attention layers. The patch size of Qwen2.5-ViT is 14.

- MoonViT-SO400M: The pre-trained vision encoder of Kimi-VL Team et al. (2025b) can encoding image with any resolution. The patch size is 14.

- ViT-UHD (ours): A naive-resolution ViT equipped with the proposed RPE and WTC modules. In this setting, the patch size in ViT-UHD is set to 10, and two WTC modules with content-adaptive pooling are inserted at the 4th and 18th layers of the original ViT, which is initialized from MoonViT-SO400M. Model details are detailed in Sec.3.1.

**A.1.1.2 Training Recipe**

**Data Curation.** We use a unified data setting for all MLLMs, consisting of two stages: (1) Stage 1 uses the LLaVA-Pretrain-558K Guo et al. (2024) dataset, which includes image-text pairs of coarse image caption. (2) Stage 2 adopts the SFT-858K Zhang et al. (2024) dataset, covering diverse instruction-following data and pure text corpus. For ViT-UHD, we introduce an additional ViT pre-alignment stage prior to the two-stage training to address the perturbations in feature modeling introduced by PVC. This pre-alignment data consists of about 4M samples, primarily sourced from a combination of image-text pairs and OCR caption datasets. Specifically, it includes data from LLaVA-Recap-118K, LLaVA-Recap-558K, and LLaVA-Recap-CC3M for image-text alignment, as well as SynthdoG-EN/ZH and UReader-TR-Processed for OCR-related tasks. Please refer to Tab. 6 for more details.

**Training Setting.** All models follow the same two-stage training setup. (1) In Stage 1, the projector is updated with learning rates 2e-4, while the ViT and LLM are frozen. (2) In Stage 2, we fine-tune the parameters of the entire model. The ViT and projector are trained with a learning rate of 1e-5,

and the LLM is optimized with a learning rate of 2e-5. For the ViT-UHD pre-alignment, we only train the ViT-UHD and projector with learning rates of 1e-5 and 2e-4, respectively. Pre-alignment is necessary. Since the RPE and WTC modules are directly integrated into the ViT architecture, they inherently alter the original encoding flow and disrupt the pre-trained feature distribution. This pre-alignment stage serves to adapt the modified network to these structural changes, ensuring that ViT-UHD retains its fundamental representational capacity before proceeding with downstream training.

### A.1.1.3 Experiment Setting

**Benchmarks.** We evaluate their performance across 6 representative benchmarks covering general perception, visual reasoning, and knowledge understanding: MMB, SEED-Bench (Image), SQA, HallusionBench, AI2D, and MMStar.

**Evaluation Metrics.** (1) We report the average accuracy across all 6 benchmarks as the main performance indicator in Fig. 1(a) and Tab. 5. (2) For efficiency evaluation, we measure the Time-to-First-Token (TTFT) latency under three input resolutions: $512 \times 512$, $768 \times 768$, and $1024 \times 1024$. TTFT includes both the ViT encoding latency and the prefilling latency of visual token in Qwen2-7B, based on the actual number of visual tokens generated by each encoder. We compute the TTFT in a unified setup using FlashAttention-2 and bf16 precision on one single 80G-A100 GPU. (3) Token counts are annotated alongside each input resolution.

### A.1.1.4 Results and Analysis

As shown in Fig. 1(a) and Tab 5, ViT-UHD achieves the best trade-off between accuracy and latency. It achieves competitive performance with other models in average accuracy while being $2.4\times$ faster than MoonViT-SO400M and $1.9\times$ faster than Qwen2.5-ViT at $1024 \times 1024$ resolution.

| Model | Resolution | TTFT (ms)↓ | Avg↑ | Benchmark | | | | | |
|---|---|---|---|---|---|---|---|---|---|
| | | | | MMB | SEED-Image | HallusionBench | SQA | AI2D | MMStar |
| SigLIP2-SO400M | 512 | 116 | 64.2 | 73.5 | 73.8 | 35.0 | 77.8 | 76.1 | 49.1 |
| Qwen2.5-ViT | 512 | 84 | 66.0 | 76.7 | 72.4 | 37.5 | 78.8 | 78.6 | 51.8 |
| | 768 | 138 | 66.7 | 77.1 | 72.9 | 38.4 | 79.2 | 79.2 | 53.5 |
| | 1024 | 225 | 66.3 | 76.0 | 72.4 | 38.2 | 78.9 | 79.2 | 53.1 |
| MoonViT-SO400M | 512 | 61 | 66.3 | 75.1 | 73.9 | 37.4 | 80.6 | 78.5 | 52.3 |
| | 768 | 146 | 67.1 | 75.5 | 74.4 | 39.5 | 80.8 | 78.7 | 53.5 |
| | 1024 | 296 | 66.9 | 75.0 | 74.5 | 38.8 | 81.1 | 78.6 | 53.5 |
| ViT-UHD | 512 | 55 | 66.8 | 77.2 | 73.5 | 37.6 | 80.8 | 78.5 | 53.1 |
| | 768 | 76 | 67.4 | 77.6 | 74.4 | 38.6 | 81.2 | 79.2 | 53.5 |
| | 1024 | 121 | 67.5 | 77.6 | 74.6 | 39.1 | 80.4 | 79.5 | 53.7 |

Table 5: Comprehensive comparison across different vision encoders under a unified Qwen2-7B LLM.

### A.1.2 COMPARISON OF LATENCY-PERFORMANCE ACROSS STATE-OF-THE-ART MLLMS

### A.1.2.1 Counterparts

We select four open-source SoTA MLLMs with our LLaVA-UHD-v3 to conduct our efficiency comparison. Among our candidates, Qwen2-VL-7B and proposed LLaVA-UHD-v3 are global naive-resolution encoding methods, and InternVL2-8B, LLaVA-OneVision-7B, and MiniCPM-V2.6-7B are slice-based methods. We also add a SoTA close-source MLLM GPT-4o-mini for comprehensive comparision.

**Overall architecture of LLaVA-UHD v3**. We employ ViT-UHD as the vision encoder, coupled with an MLP projector and Qwen2-7B as the LLM. The ViT-UHD is configured with a patch size of 10, and 3 WTC layers are integrated at the 4th, 18th, and 27th layers of the original ViT. These WTC layers utilize content-adaptive pooling for the first two positions and pixel-unshuffle for the final position, respectively.

#### A.1.2.2 Experiment Setting

**Benchmarks.** We evaluate our model on a comprehensive suite of 15 multi-modal benchmarks, which are divided into four categories: (1) General benchmarks including MME, MMB, SEED, and MMStar. (2) Knowledge benchmarks such as SQA, MMMU, AI2D and MathVista. (3) OCR&Chart benchmarks consist of OCRBench, TextVQA, DocVQA and ChartQA. (4) Visual reasoning benchmarks like HallusionBench, RealWorldQA and CV-Bench.

**Evaluation Metrics.** (1) We report Time-to-First-Token (TTFT) as the primary latency metric, measured under a unified setup using FlashAttention-2 and bf16 precision. All models are tested using a fixed input resolution $1344 \times 1344$, ensuring consistent input conditions for fair architectural comparison. (2) We report accuracy of each benchmark for comprehensive comparison.

#### A.1.2.3 Results and Analysis

As shown in Fig. 1(b) (top), LLaVA-UHD v3 achieves the lowest latency among all models, with a TTFT of 153.8ms, representing a 49% reduction compared to Qwen2-VL-7B and a 10% reduction over MiniCPM-V2.6-7B. These results highlight that our approach achieves higher efficiency than slice-based encoding methods, while preserving the global modeling capability of naive-resolution encoding, achieving competitive performance with Qwen2-VL shown in Fig. 1(b) (bottom). Note that, for state-of-the-art performance, we additionally apply simple and reproducible test-time augmentations to LLaVA-UHD v3. Specifically, we design task-specific prompt refinements for different benchmarks and proportionally upscale original images in tasks requiring fine-grained text or detail recognition. These enhancements are only applied to our model and are not used for baseline results reported from prior papers. We highlight them here to ensure transparency, and the detailed prompt adjustments, as well as up-scaling ratios can be found in our released evaluation code.

### A.2 Implementation Details of Pilot Experiment

#### A.2.1 Training Setting of Pilot Experiment

#### A.2.1.1 Model Configuration

For both global naive-resolution encoding (GNE) and slice-based methods (SBE), we adopt a unified MLLM architecture with SigLIP2-SO400M (Tschannen et al. (2025)) as the vision encoder, a pixel-unshuffle (Dong et al. (2024)) projector, and Qwen2-7B (Team (2024b)) as the LLM.

#### A.2.1.2 Training Recipe

**Data Curation.** We use a unified data setting for all models, consisting of two stages: (1) Stage 1 employs the 4.3M-scale dataset as defined in the overall Stage 1 setting as shown in Tab. 1, which includes image-text pairs of coarse image caption. (2) Stage 2 adopts the SFT-858K Zhang et al. (2024) dataset, covering diverse instruction-following data and pure text corpus.

**Training Setting.** All models follow the same two-stage training setup. (1) In Stage 1, the projector is updated with learning rates 2e-4 and the vision encoder with 1e-5, while the LLM is frozen. (2) In Stage 2, we fine-tune the parameters of the entire model. The ViT and projector are trained with a learning rate of 1e-5, and the LLM is optimized with a learning rate of 2e-5.

**Hyper-parameters of visual encoding.** For GNE methods, images are preserved at their original resolution. For SBE approach, we following Guo et al. (2024); Liu et al. (2024c). Specially, in the stage 1, vision encoder only encoding a image thumbnail of $336 \times 336$ resolution. In the stage 2, it partitions and encodes the image of naive resolution into $336 \times 336$ patches at any aspect ratio, with a maximum of 9 slices per image. The input resolution is set to any value up to $1000 \times 1008$, ensuring that the experimental results are not biased by the resolution range used during training. We use the image of naive resolution for model inference.

#### A.2.2 Construction details of ShapeGrid

**ShapeGrid subset.** Each ShapeGrid image is constructed from a template pool, where each template ($336 \times 336$ pixels) contains a randomly sampled geometric shape. The pool covers 9 shape categories (*e.g.*, triangle, square, pentagram) and 7 color variants (*e.g.*, red, yellow, blue), with random perturbations in scale and position to increase variability. Final images (about 4000 samples)

are assembled by sampling templates without replacement and arranging them into 5 predefined layouts (1×2, 2×3, 2×2, and their transpose), deliberately aligned with the slicing configurations of sliced-based encoding. Based on these images, we define four spatial perception tasks: relative distance, relative position, relative area and counting, shown in Fig. 2(a).

**Sudoku subset.** To ensure that the behavior of the both global naive-resolution encoding and slice-based encoding methods more accurately reflects their ability in spatial positional understanding, we extend the original location subset into a Sudoku-style location subset. This subset can be regarded as a refinement of the original location task, in which directional localization is explicitly isolated as the primary factor for probing spatial perception. Each Sudoku image is arranged in a 3×3 grid, with the center cell fixed as a red pentagram serving as the reference anchor, and the remaining eight cells randomly selected from the template pool, including additional real-world categories (*e.g.*, bear, car, plane). This design produces 8,000 images, each paired with a query asking for the relative direction of the target object with respect to the central pentagram, shown in Fig. 3(a).

### A.2.3 SUDOKU ACCURACY ON COMMERCIAL OPEN MLLMS

To verify whether the performance of global naive-resolution visual encoding and slice-based encoding on the Sudoku subset exhibits consistent patterns observed in the pilot experiment, we further evaluate the widely discussed approaches, like Qwen2.5-VL representing GNE and MiniCPM-o 2.6 representing SBE on the Sudoku subset. To increase the task complexity, we reformulate the problem by replacing relative position queries with absolute position queries, wherein the target may appear at any location within the 3×3 grid, and the model is required to predict its precise row and column indices. The results are shown in Fig. 6. It can be seen that Qwen2.5-VL achieves consistently high accuracy across all positions in the Sudoku subset, whereas MiniCPM-o 2.6 exhibits biases similar to those observed in Sec. 2.2, with significantly lower accuracy in the top and right positions. This observation is consistent with our pilot experiment results and further demonstrates that the inherent limitations of slice-based encoding cannot be mitigated even with large-scale data training. For instance, although MiniCPM-o 2.6 7B and Qwen2.5-VL-7B achieve nearly equivalent overall performance on OpenCompass (70.2 vs. 70.9), these structural weaknesses persist.

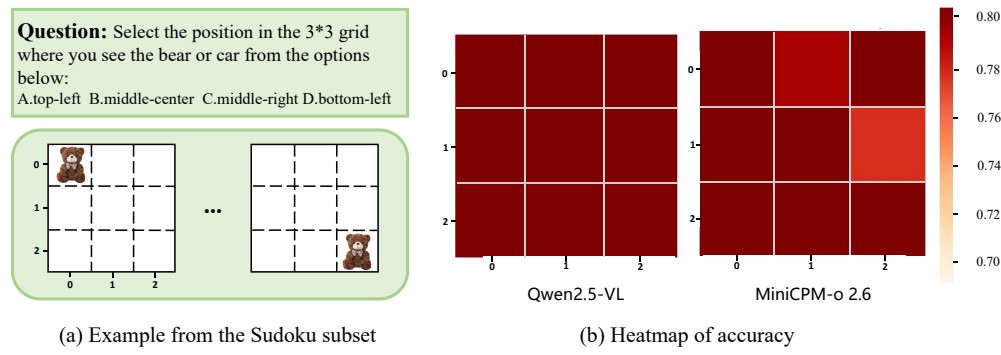

(a) Example from the Sudoku subset  (b) Heatmap of accuracy

Figure 6: Accuracy heatmap of MiniCPM-o 2.6 and Qwen2.5-VL on the Sudoku subset.

### A.3 OVERALL TRAINING RECIPE

#### A.3.1 DATA CURATION

We train our proposed LLaVA-UHD-v3 using a three-stage data pipeline, as shown in Tab. 6.

**Stage 1: Vision-language alignment**. We collect 4.3M general-purpose image-caption pairs from LLaVA-Pretrain Liu et al. (2023c), LLaVA-OneVision-Pretrain Li et al. (2024a), and other sources to facilitate vision-language alignment and vision encoder pretraining.

**Stage 2: Joint multi-modal pretraining**. We construct a 5M dataset comprising coarse-grained OCR caption data, interleaved multimodal samples, and a subset of replayed Stage-1 data to further unify the visual and language representations.

| Stage | Dataset | Type | #Samples | Total |
|-------|---------|------|----------|-------|
| **Stage 1** | LLaVA-Pretrain | Image-Text Pairs | 558K | 4.3M |
| | LLaVA-Recap-118K | Image-Text pairs | 118K | |
| | LLaVA-Recap-558K | Image-Text pairs | 558K | |
| | LLaVA-Recap-CC3M | Image-Text pairs | 2.8M | |
| | SynthdoG-EN/ZH | OCR Caption | 200K | |
| | UReader-TR-Processed | OCR Caption | 101K | |
| **Stage 2** | Captioned OCR Data | OCR | 832K | 5M |
| | Coarse-grained Doc& Chart Data | Doc, Chart | 790K | |
| | Obelics | Interleaved | 658K | |
| | Pure Text Data | Text | 1M | |
| | Replayed Data From Stage-1 | Image-Text Pairs | 1.2M | |
| | Allava-Laion | Detailed Image Caption | 240K | |
| | Wit | Detailed Image Caption | 280K | |
| **Stage 3** | MammothVL-12M | Instruction-Following | 12M | 13.3M |
| | LLaVA-SFT-858K | Instruction-Following | 858K | |
| | Video-R1-CoT Feng et al. (2025) | Reasoning | 80K | |
| | SiCOG Zhang et al. (2025) | Detailed Image Caption, Reasoning | 70K | |
| | PixMo | OCR, Chart, Doc | 310K | |

Table 6: Composition of training data across different stages.

**Stage 3: Supervised fine-tuning**. We build our SFT data pool based on MammothVL-12MGuo et al. (2025), and enrich it with fine-grained charts, documents, and complex reasoning samples to enhance the model's instruction-following capabilities.

### A.3.2 DETAILS OF TRAINING SETTING

Based on the Tab. 1, we employ a cosine learning rate schedule with a 3% warm-up phase at the beginning of each stage, and optimize the model using AdamW with $\beta = 0.9, 0.95$ and a weight decay of 0.

### A.4 TRAINING SETTING FOR MODULE ABLATIONS

#### A.4.1 Model Configuration

For the baseline, we adopt SigLIP2-SO-400M as the vision encoder with a WTC (using pixel-unshuffle) module appended in the last ViT layer , an MLP as the projector, and Qwen2-7B as the LLM. For the PVC incremental model, we insert two WTC layers into the vision encoder, which can be implemented as average pooling, content-adaptive pooling, or pixel unshuffle. For the RPE-equipped model, we add two WTC layers with content-adaptive pooling and a resized patch embedding layer with patch size 8.

#### A.4.2 Training Recipe

The training recipe is the same as that described in Appendix A.1.1. We also give the details as following.

**Data Curation.** We use a unified data setting for both of baseline and module incremental development, consisting of two stages: (1) Stage 1 uses the LLaVA-Pretrain-558K Guo et al. (2024) dataset (2) Stage 2 adopts the SFT-858K Zhang et al. (2024) dataset. Note that, for module incremental development, like inserting WTC in the 4/18-th layer, we introduce an additional ViT pre-alignment stage prior to the two-stage training to address the perturbations in feature modeling introduced by PVC.

**Training Setting.** All experiments follow the same two-stage training setup. (1) In Stage 1, the projector is updated with learning rates 2e-4, while the ViT and LLM are frozen. (2) In Stage 2, we fine-tune the parameters of the entire model. The ViT and projector are trained with a learning rate of 1e-5, and the LLM is optimized with a learning rate of 2e-5. For the pre-alignment, we only train the transformed ViT and projector with learning rates of 1e-5 and 2e-4, respectively.

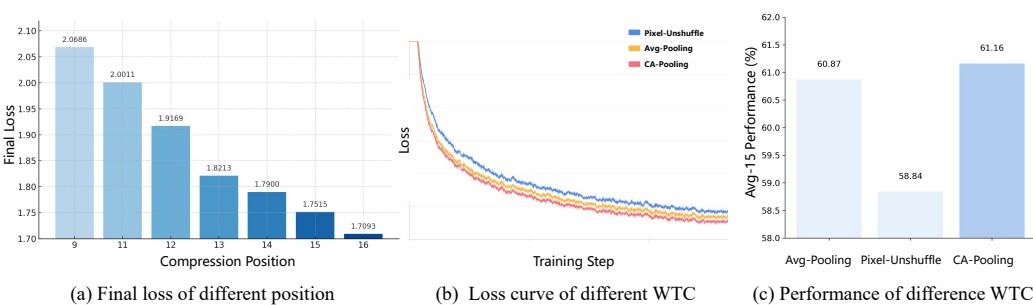

(a) Final loss of different position    (b) Loss curve of different WTC    (c) Performance of difference WTC

Figure 7: Evaluations on hyper-parameters of WTC type and layer position. All the ablation is conducted on SigLIP2 at 1024×1024 with patch size 16. (a) illustrates the final loss when using pixel-unshuffle in different layer position. (b) shows the training loss curve when using different WTC. (c) gives the performance comparison of different WTC.

### A.5 ADDITIONAL ABLATION STUDY

We conduct additional experiment for comprehensive analysis on our proposed methods.

#### A.5.1 EXPERIMENT ON DIFFERENT WTC

As shown in Fig. 7(a), when integrating WTC with pixel-unshuffle into the ViT architecture, we observe that placing WTC at lower layers leads to greater disruption of training convergence. This demonstrates the point discussed in Appendix A.1.1, namely that inserting additional modules directly into the ViT disturbs the pre-trained feature modeling flow, thereby necessitating an extra pre-alignment stage to stabilize the network. Moreover, in Fig. 7(b), we find that avg-pooling, despite being parameter-free, achieves better convergence stability than pixel-unshuffle when inserted at the same positions (*i.e.*, 4-th and 18-th layers). Finally, our content-adaptive pooling further enhances convergence stability and efficiency, shown in Fig. 7(b), yielding faster convergence and achieving superior average performance across 15 benchmarks, as illustrated in Fig. 7(c).

#### A.5.2 EXPERIMENT ON DIFFERENT VISION ENCODER

To examine whether the proposed Progressive Visual Compression (PVC) method generalizes beyond a single ViT, we conducted transferability experiments on both MoonViT and SigLIP2. In each case, we integrated PVC, including RPE and WTC into the backbone without modifying other components. The results show consistent improvements: PVC reduces visual token count and time-to-first-token (TTFT), while improving average accuracy. These findings demonstrate that PVC is readily portable and yields reliable benefits across ViT architectures.

| Backbone | PVC | Patch Size | Layer | Tokens↓ | TTFT (ms)↓ | Avg↑ |
|---|---|---|---|---|---|---|
| MoonViT | ✗ | 14 | None | 1332 | 386.17 | 66.17 |
| MoonViT | ✓ | 10 | (4, 18) | **655** | 186.24 | **66.33** |
| SigLIP2 | ✗ | 16 | None | 1024 | 233.01 | 61.94 |
| SigLIP2 | ✓ | 8 | (4, 18) | **256** | 159.69 | **64.71** |

Table 7: Performance and efficiency when integrating PVC on MoonViT and SigLIP2 vision encoder.

### A.6 FUTURE WORK AND LIMITATION

#### A.6.1 LIMITATIONS

**MLLM performance gains do not imply fully optimized ViT pretraining.** We further investigate whether the vision encoder in LLaVA-UHD v3 has reached its full potential by re-running the experi-

ment in Fig. 1(a) using the trained ViT-UHD without applying the pre-alignment stage. Interestingly, we find that its relative performance does not improve significantly, and in some benchmarks even degrades, despite LLaVA-UHD v3 achieving state-of-the-art performance at the system level. This suggests that the current MLLM training pipeline alone may be insufficient to fully optimize the ViT, leaving its visual representation learning far from saturated.

**Still faces quadratic complexity explosion.** Although our method effectively controls computational overhead, it does not fundamentally overcome the quadratic complexity of self-attention. When the input resolution grows exceedingly large (*e.g.*, beyond 4K), the model still encounters substantial computational costs and long inference latency.

### A.6.2 FUTURE WORK

In future work, we will focus on MLLM-centric pretraining of vision encoders, aiming to build a stronger visual backbone for more effective vision-language alignment and reasoning. To reduce training costs, we plan to initialize from existing pretrained models and further adapt them through targeted post-training, while progressively replacing quadratic-complexity operators with linear-complexity counterparts to achieve scalable and efficient multimodal modeling.

## A.7 CASE STUDY

### A.7.1 COMPARISON WITH BASELINES

**Comparison of LLaVA-UHD-v3 with Qwen2-VL and MiniCPM-V-2.6**

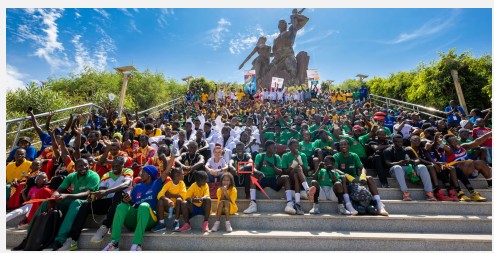

*Input image*        *Key information highlighted*

**Question:** What is the color of the band on the arm of the person sitting to the left of the white girl in the picture?

**Qwen2-VL Answer(✗):** The band on the arm of the person sitting to the left of the white girl in the picture is **red**.

**MiniCPM-V-2.6 Answer(✗):** The color of the band on the arm of the person sitting to the left of the white girl is **red**.

**LLaVA-UHD-v3 Answer(✓):** The person sitting to the left of the white girl in the picture is wearing a **blue** band on their arm. This detail stands out among the crowd, which appears to be diverse and colorful, with many people wearing various colored shirts and accessories.

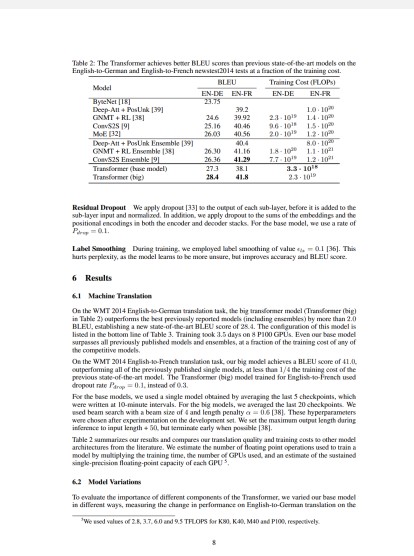 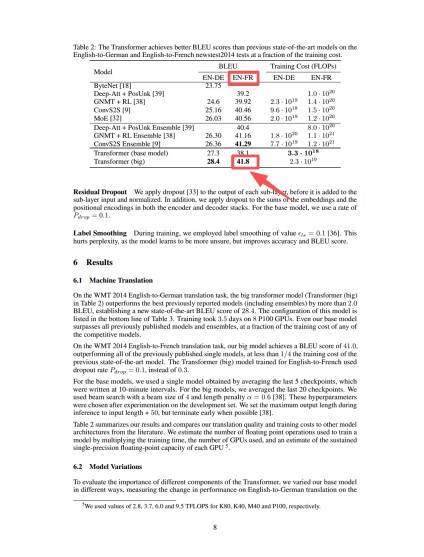

*Input image*        *Key information highlighted*

**Original Question:** Based on the information in the table, explain which method performs best in terms of BLEU on the English-to-French translation task.

**Qwen2-VL Answer(✗):** The Transformer (big) model performs best in terms of BLEU on the English-to-French translation task, achieving a score of **41.0**.

**MiniCPM-V-2.6 Answer(✗):** The best method for the English-to-French translation task, as per the table, is the Transformer (big) model, which achieves a BLEU score of **41.0** on the WMT 2014 test set.

**LLaVA-UHD-v3 Answer(✓):** The method that performs best in terms of BLEU on the English-to-French translation task is the "Transformer (big)" with a BLEU score of **41.8**.

### A.7.2 SHOWCASE OF OUR MODEL'S CAPABILITIES

**Image Captioning**

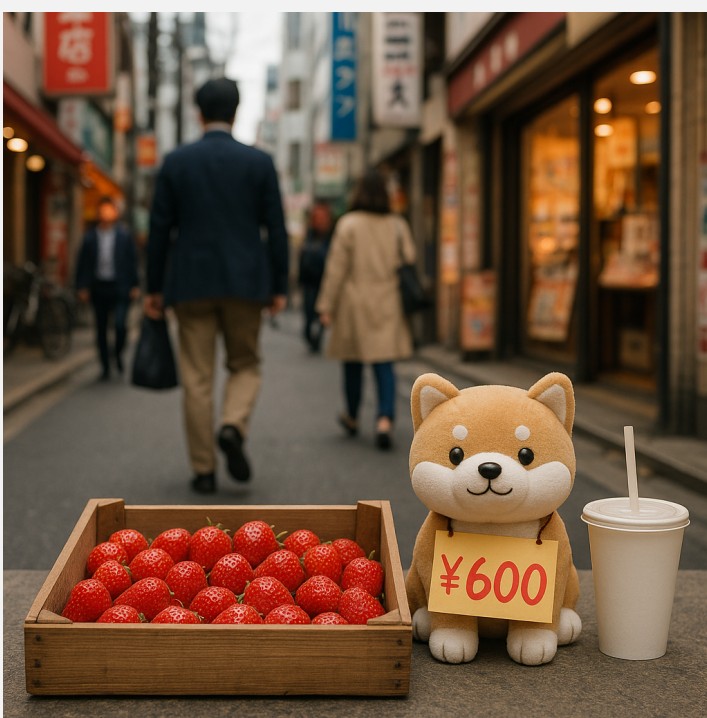

*Input image*

**Question:** What is shown in this image?

**LLaVA-UHD-v3:** The image shows a charming street scene in Japan. In the foreground, there's a wooden crate filled with fresh strawberries and a stuffed animal dog holding a sign that says "¥600". Next to the dog is a white cup with a straw.

In the background, you can see people walking down the street, including a man in a blue suit carrying a black bag and a woman in a tan coat with blue jeans. The buildings lining the street have various signs, some of which appear to be in Japanese.

This scene captures a typical day on a Japanese street, blending everyday life with a touch of whimsy from the stuffed animal and its strawberry display.

## OCR Case

| Metal | Reaction | Electrode Potential (V) |
|-------|----------|-------------------------|
| Gold | $Au^+ + e^- = Au$ | + 1.692 |
| Silver | $Ag^+ + e^- = Ag$ | +0.7996 |
| Copper | $Cu^{2+} + 2e^- = Cu$ | +0.342 |
| Iron | $Fe^{3+} + 3e^- = Fe$ | -0.037 |
| Lead | $Pb^{2+} + 2e^- = Pb$ | -0.126 |
| Nickel | $Ni^{2+} + 2e^- = Ni$ | -0.257 |
| Cadmium | $Cd^{2+} + 2e^- = Cd$ | -0.403 |
| Iron | $Fe^{2+} + 2e^- = Fe$ | -0.447 |
| Zinc | $Zn^{2+} + 2e^- = Zn$ | -0.762 |
| Aluminum | $Al^{3+} + 3e^- = Al$ | -1.662 |

*Input image*

**Prompt:** Convert to markdown.

**LLaVA-UHD-v3:**

```
| Metal      | Reaction                       | Electrode Potential (V) |
|------------|--------------------------------|-------------------------|
| Gold       | Au+ + e-  = Au                 | +1.692                  |
| Silver     | Ag+ + e-  = Ag                 | +0.7996                 |
| Copper     | Cu2+ + 2e- = Cu                | +0.342                  |
| Iron       | Fe3+ + 3e- = Fe                | -0.037                  |
| Lead       | Pb2+ + 2e- = Pb                | -0.126                  |
| Nickel     | Ni2+ + 2e- = Ni                | -0.257                  |
| Cadmium    | Cd2+ + 2e- = Cd                | -0.403                  |
| Iron       | Fe2+ + 2e- = Fe                | -0.447                  |
| Zinc       | Zn2+ + 2e- = Zn                | -0.762                  |
| Aluminum   | Al3+ + 3e- = Al                | -1.662                  |
```

## Counting Case

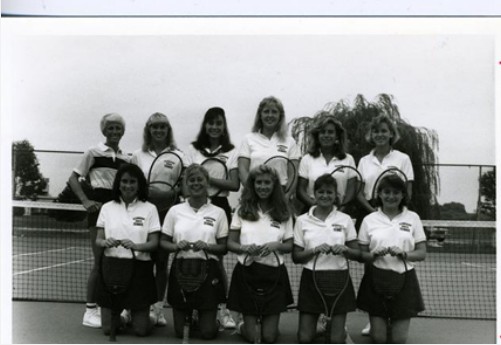

*Input image*

**Question:** How many tennis rackets are in the image?

**Options:** A.9 B.10 C.13 D.11 E.12 F.0

**LLaVA-UHD-v3:** D.11

**High-Resolution Detail Case**

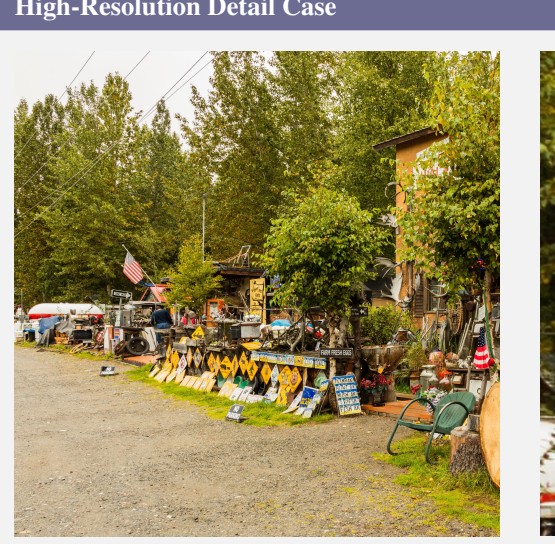
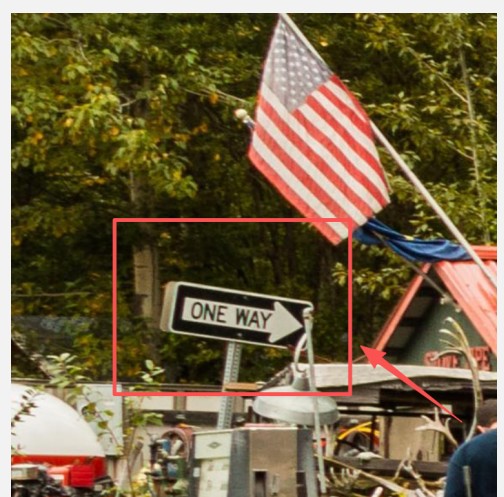

*High-Resolution (4K) Input*                    *Key information highlighted*

**Question:** Below the flag on the left side of the image, there is a road sign. What is written on it?

**LLaVA-UHD-v3:** The road sign below the flag on the left side of the image says "ONE WAY".

