# OpenReview forum: "LLaVA-UHD v3: Progressive Visual Compression for Efficient Naive-Resolution Encoding in MLLMs"
_ICLR.cc/2026/Conference — ICLR 2026 Conference Withdrawn Submission_

### Official Review · Reviewer_13xw · 2025-10-27

**Soundness:** 3
**Presentation:** 2
**Contribution:** 2
**Rating:** 4
**Confidence:** 4

**Summary:**

This paper introduces LLaVA-UHD v3, a multimodal large language model designed around Progressive Visual Compression (PVC) to enable computationally efficient, naive-resolution encoding within standard Vision Transformers (ViTs). PVC consists of two main modules: Refined Patch Embedding (RPE), enabling flexible patch size adjustment via a principled weight transformation, and Windowed Token Compression (WTC), which reduces token sequences hierarchically during encoding through pooling operations (both average and content-adaptive). The paper presents controlled pilot experiments comparing global naive-resolution versus slice-based encoding for vision-language understanding, shows the architectural mechanism of PVC, and provides extensive benchmarks, ablations, and case studies. Results indicate that the proposed approach achieves comparable or superior accuracy with significantly lower time-to-first-token (TTFT) than baseline MLLMs across a wide range of tasks and input resolutions.

**Strengths:**

1. Comprehensive Empirical Evaluation: The paper evaluates LLaVA-UHD v3 extensively across a diverse set of vision-language benchmarks (Tables 2 & 3), including general, OCR, chart, reasoning, and knowledge tasks, and provides ablation studies (Table 4, Figure 5) showing the effect of individual components. The comparison baseline covers both academic and commercial models, ensuring relevance.

2. Detailed Analysis of Encoding Paradigms: The pilot experiments (Section 2, Figure 2 and Figure 3) give insight into the effects of global naive-resolution versus slice-based encoding on both semantic and spatial perception. The authors use synthetic benchmarks and attention maps to expose systematic flaws in slice-based approaches, and clearly motivate the need for efficient global encoders.

**Weaknesses:**

1. **Missing Discussion of Several Directly Related Recent Works**: Several directly relevant works are not discussed or cited in the main text, including FlashSloth, NVILA, and Scaling Vision Pre-Training to 4K Resolution. The lack of comparison or disambiguation with these highly related methods, which focus on efficient visual compression and high-resolution encoding for MLLMs, is a serious omission that hinders positioning in the broader landscape. This must be rectified in revision, particularly in the Related Work section, with explicit qualitative and (where feasible) quantitative comparison.

2. **Limited Theoretical Analysis of Compression-Performance Trade-off**: While the empirical results are strong, the paper lacks a more rigorous theoretical discussion on why PVC’s sequence-length reduction via content-adaptive pooling preserves relevant information, or an explicit quantification of the information-accuracy trade-off. In Section 3.1.2, the adaptive pooling mechanism is heuristically motivated, but no theoretical guarantees or limitations are provided. For example, what are the expected bounds on accuracy degradation as patch size decreases or compression stages increase?

3. **Compression Overhead is Not Fundamentally Removed**: As outlined in Limitations (Appendix A.6.1), the method does not fundamentally overcome the quadratic complexity of self-attention in ViTs. When scaling to ultra-high resolutions (e.g., 4K and above), computational overhead still grows rapidly, and the efficiency gains saturate as indicated in Figure 5. This should be highlighted more openly in the main narrative and not just the Appendix.

4. **Potential Biases Introduced by Training and Evaluation Details**: In Appendix A.1.2.3, the use of task- and benchmark-adapted prompt augmentations for test-time efficiency/performance is only mentioned briefly. These augmentations can confound the fairness of direct comparison with prior baselines if not standardized across all models. Furthermore, such prompt engineering may artificially inflate scores on certain benchmarks.

5. **Opaque Patch Embedding Weight Transformation**: The RPE module relies on the least-squares pseudo-inverse transformation (Section 3.1.1, Equation), but the process for computing the transformation matrix $B$ and corresponding pseudo-inverse is only sketched. There is no discussion of the practical stability, conditioning, or bias that could arise when applying the operation to pretrained weights from diverse ViTs. Further experimental or analytical examination is needed, especially for porting to non-standard architectures.

6. **Results Reporting and Benchmark Transparency**: Some reproduced or externally-sourced scores in Tables 2/3 (noted with *) are not clearly cross-validated. Disparity in evaluation pipelines (especially concerning use of augmentations or upscaling) can bias the ranking.

7. **Explanatory Depth in Mathematical Details**: While the overall math is accurate, some details around the learnable aggregation (content-adaptive pooling) in Section 3.1.2 could benefit from additional justification — specifically, when/why the softmax aggregation meaningfully outperforms simple averaging, and whether this introduces instability when training from scratch.

8. **Still Lacking Truly Novel Architectural Breakthrough**: Although PVC offers a creative combination and principled modularization of patch resizing and windowed token aggregation, the core technical ingredients (average pooling, learnable MLP attention, patchwise linear ops) are evolutionary, not fundamentally new. The main contribution lies in the systematic unification and empirical demonstration, but the paper should be clearer on this point.

9. **Minor Clarity Issues**: Some writing—especially in the experimental and ablation settings—could be made more concise and accessible. Key design choices (e.g., number/position of WTC layers in Table 4, rationale for selected patch sizes) are occasionally justified post-hoc and could benefit from stronger up-front motivation.

**Reference**

[1] Tong, B., Lai, B., Zhou, Y. (2025): FlashSloth: Lightning Multimodal Large Language Models via Embedded Visual Compression. Directly addresses embedded visual compression for efficient MLLMs, paralleling the PVC approach, but is not discussed. Should be cited and compared in Section 5 and included as a baseline for quantitative assessment.

[2] Liu, Z., Zhu, L., Shi, B. (2025): NVILA: Efficient Frontier Visual Language Models. Proposes efficient vision-language models prioritizing both compression and accuracy—highly relevant for contextualizing LLaVA-UHD v3. Should be cited in Related Works and compared in Section 4 results.

[3] Shi, B., Li, B., Cai, H. (2025): Scaling Vision Pre-Training to 4K Resolution. This work addresses large-scale pretraining for high resolution, directly aligning with the main challenge LLaVA-UHD v3 addresses. Should appear in Related Works.

[4] Zhu, C., Wang, T., Zhang, W. (2025): LLaVA-3D: A Simple yet Effective Pathway to Empowering LMMs with 3D Capabilities. Although focused on 3D, offers alternate perspectives on efficient input tokenization; merits mention in Related Works.

[5] Li, S., Hu, Y., Ning, X. (2025): MBQ: Modality-Balanced Quantization for Large Vision-Language Models. Pertinent to the focus on model size and efficient multimodal processing; should be discussed briefly in Related Works.

[6] Wang, J., Liu, Z., Rao, Y. (2025): SparseMM: Head Sparsity Emerges from Visual Concept Responses in MLLMs. Offers perspective on sparsification and efficient attention, closely linked to iterative compression; Related Works section.

[7] Zheng, C., Zhang, J., Salehi, M. (2025): One Trajectory, One Token: Grounded Video Tokenization via Panoptic Sub-object Trajectory. While focused on video, their approach for reducing token redundancy is relevant and should be briefly referenced.

[8] Li, H., Zhang, Y., Guo, L. (2025): Breaking the Encoder Barrier for Seamless Video-Language Understanding. Proposes NOVA, an encoder-free model for efficient video-language modeling. Can be briefly discussed when outlining alternative architectures.

[9] Yang, Z., Luo, X., Han, D. (2025): Mitigating Hallucinations in Large Vision-Language Models via DPO: On-Policy Data Hold the Key. This paper’s discussion on robustness relates to the claim of higher reliability and efficiency for LLaVA-UHD v3; a short mention is warranted in Section 5.

**Questions:**

Please see the weaknesses. I will consider raising my score if the authors can resolve my concerns well.

---

### Official Review · Reviewer_obPv · 2025-10-27

**Soundness:** 2
**Presentation:** 3
**Contribution:** 2
**Rating:** 4
**Confidence:** 3

**Summary:**

This paper presents LLaVA-UHD v3, a multimodal large language model (MLLM) designed to enable efficient naive-resolution visual encoding by introducing a Progressive Visual Compression (PVC) framework. The authors first conduct controlled experiments comparing global naive-resolution encoding with slice-based encoding and show that global encoding yields better semantic understanding and spatial reasoning. Then, they propose PVC to address the computational cost induced by global encoding. PVC can be seamlessly integrated into a pretrained Vision Transformer to form ViT-UHD. PVC consists of two components: Refined Patch Embedding (RPE), which increases visual granularity through flexible patch-size scaling while preserving pretrained weights, and Windowed Token Compression (WTC), which progressively reduces token length across layers using lightweight, learnable local aggregation. Built on this design, ViT-UHD surpasses MoonViT and Qwen2.5-ViT within the same MLLM framework, while achieving a substantial reduction in time-to-first-token (TTFT). LLaVA-UHD v3 attains performance competitive with some mainstream MLLMs (e.g., Qwen2-VL) and simultaneously delivers notable TTFT efficiency gains.

**Strengths:**

- ViT-UHD follows a clear and logical design: it first enhances visual granularity through Refined Patch Embedding (RPE), then progressively reduces token count via Windowed Token Compression (WTC). The framework is modular and can be seamlessly applied to pretrained ViTs without re-training from scratch.
- ViT-UHD demonstrates promising results, outperforming mainstream ViT-based encoders in both downstream accuracy and efficiency.
- The paper is generally well written, well structured, and easy to follow.

**Weaknesses:**

- The main experiments rely on Qwen2-7B as the LLM and compare against relatively outdated MLLMs. As a result, the potential of this work to advance the frontier of MLLMs remain unclear. Stronger and more recent backbones/baselines (e.g., Qwen3, Qwen2.5-VL, InternVL3.5) should be included to better contextualize the contributions.
- The pilot study mainly shows that global naive-resolution encoding outperforms slice-based encoding in capability, but lacks an analysis of computational overhead. Without quantifying this cost, the motivation for a more efficient global encoding method (such as PVC) feels incomplete.
- The paper lacks a thorough examination of the effect of RPE patch size.
- The paper does not comapre against recent visual token compression methods [1,2,3].

[1] An Image is Worth 1/2 Tokens After Layer 2: Plug-and-Play Inference Acceleration for Large Vision-Language Models.

[2] VisionZip: Longer is Better but Not Necessary in Vision Language Models.

[3] SparseVLM: Visual Token Sparsification for Efficient Vision-Language Model Inference.

**Questions:**

Please refer to the **Weaknesses**

---

### Official Review · Reviewer_kLee · 2025-11-01

**Soundness:** 3
**Presentation:** 3
**Contribution:** 2
**Rating:** 4
**Confidence:** 5

**Summary:**

This paper addresses the high computational cost of global naive-resolution vision transformer in MLLMs.

The authors first present a pilot study comparing global naive-resolution encoding with slice-based encoding, using a new synthetic benchmark called ShapeGrid. They find that GNE provides superior spatial and semantic understanding, while SBE suffers from a "cross-shaped bias".

To mitigate the high cost of GNE, the authors propose LLaVA-UHD v3, a model built on a Progressive Visual Compression framework, with two components: a Refined Patch Embedding, which uses a weight transformation technique to allow for finer-grained patches while leveraging pretrained ViT weights, and Windowed Token Compression, which inserts lightweight, content-adaptive pooling layers hierarchically within the ViT to progressively merge tokens and reduce sequence length.

Experiments show that the resulting model, ViT-UHD, achieves significant efficiency gains (e.g., 2.4x faster TTFT than MoonViT) while maintaining competitive performance on various vision-language benchmarks.

**Strengths:**

1. The paper tackles a timely and significant problem in MLLM research. the trade-off between ViT resolution (and performance) and computational efficiency.
2. Pilot study. The pilot experiment in Sec. 2 is a valuable contribution. The analysis of GNE vs. SBE is insightful, and the finding of a "cross-shaped bias" in SBE is a clear, interesting result, though not surprising. The introduction of the ShapeGrid benchmark for this specific analysis is well-motivated.
3. The proposed PVC framework is intuitive. Combining finer-grained input tokens (via RPE) with a mechanism to intelligently reduce token count during encoding (via WTC) is a logical approach to balancing detail and efficiency.
4. The efficiency advantage is significant. The paper demonstrates impressive improvements in inference latency (TTFT) while maintaining competitive, and in some cases superior, performance against strong baselines, as is shown in Tab. 1 and Fig. 2 and 3.
5. The authors provide several ablation studies that analyze the impact of different WTC types, their positions, and the RPE module.

**Weaknesses:**

1. The novelty of the components seems incremental. The motivation of RPE is interesting, as it notices the patch size could be adjusted, but it is based on the method from Beyer et al. (2023) and is not a novel contribution itself., For WTC, the idea of progressive token merging inside a ViT is the core of hierarchical transformers (e.g., Swin Transformer). The proposed "content-adaptive pooling" (Eq. 2) is a simple, parameterized attention/gating mechanism on top of average pooling. While the combination is effective, the components themselves are well-established or simple variations of existing ideas.

2. Experimental fairness. This is the most significant concern. The ablation studies (Sec 4.1, A.1.1.2) state that an extra "ViT pre-alignment stage" using ~4M samples is applied to all PVC-based models (including the final ViT-UHD). As noted, this stage is explicitly used "to mitigate the perturbation... introduced by PVC". However, this pre-alignment step does not appear to be applied to the baseline methods in the ablations like the pixel-unshuffle model. This seems not a fair comparison. The performance gains and convergence stability attributed to the proposed WTC module may be largely due to this extra training step, which the baselines did not receive.

3. Following the point above, the paper claims that parameterized methods like pixel-unshuffle exhibit convergence challenges and cause notable performance degradation. It is unclear if these challenges are inherent to the methods themselves or are simply a result of not being given the same pre-alignment stage as the proposed method. This weakness undermines the central justification for the authors' specific WTC design.

4. The appendix notes that rerunning the training without applying the pre-alignment stage degrades performance. This confirms this stage is critical. How much of the model's final performance and stability do you attribute to this extra, computationally expensive pre-alignment step versus the architectural design of PVC itself?

**Questions:**

Please see the comments above regarding the weaknesses, which shows how each concern can be discussed and addressed in the rebuttal/revision.

Additionally, a small question: this work seems not directly based on llava-uhd v1 and v2. It is not even based on llava series. Why is it named llava-uhd v3?

---

### Note · Authors · 2025-11-20

I have read and agree with the venue's withdrawal policy on behalf of myself and my co-authors.